# Number of antenatal care utilization and associated factors among pregnant women in rural Ethiopia: Zero-inflated Poisson regression of 2019 intermediate Ethiopian Demography Health Survey

**Bisrat Misganaw Geremew[1], Yitbarek Fantahun Mariye[2], Daniel Gashaneh Belay[3], Hiwot Tezera Endale[4], Fana Kinfe Gebreegziabher[5], Habtu Kifle Negash** [3]*

1 Department of Epidemiology and Biostatistics, Institute of Public Health, College of Medicine and Health Sciences, University of Gondar, Gondar, Ethiopia, 2 Department of Obstetrics & Gynecology, School of Medicine, College of Medicine & Health Sciences, Addis Ababa University, Addis Ababa, Ethiopia, 3 Department of Anatomy, College of Medicine and Health Sciences, University of Gondar, Gondar, Ethiopia, 4 Department of Biochemistry, College of Medicine and Health Sciences, University of Gondar, Gondar, Ethiopia, 5 Department of Management, College of Business and Economics, University of Gondar, Gondar, Ethiopia

* habtuk8@gmail.com

## Abstract

### Background

About 70% of maternal fatalities (202,000) occurred in Sub-Saharan Africa alone. ANC lowers the morbidity and death rates for mothers and perinatals. The study aimed to determine the number of antenatal care and associated factors in the rural part of Ethiopia.

### Methods

We performed secondary data analysis for the 2019 intermediate EDHS, utilizing weighted data from a total of 2896.7 pregnant women. A zero-inflated Poisson regression analysis was executed using Stata version 17.0. Using the incident rate ratio and odds ratio with a 95% confidence interval, the intensity of the link and direction were shown.

### Result

From the weighted pregnant women, 1086(37.47%) used four or more antenatal care during the current pregnancy. About 848 (29.29%) women do not attend antenatal care during pregnancy. The highest percentage (29.65%) of antenatal care visits was observed from 25 to 29 age. In comparison to women without formal education, the frequency of prenatal visits was 1.1(IRR = 1.1, 95% CI: 1.0425, 1.175) and 1.2 (IRR = 1.2, 95% CI: 1.093, 1.308) times higher among women enrolled in primary, and secondary & above education, respectively. Orthodox followers had 13% (IRR = 0.87, 95% CI: .813, 0.935) and 31% (IRR = 0.69, 95% CI: .552, 0.870) times more prenatal care visits than Protestant and other religions' followers, respectively. Prenatal care was substantially more common among women living in

**Data Availability Statement:** The minimal data are within the article and its Supporting Information

files. The data used in this study are publicly available from the Demographic and Health Surveys (DHS) Program and can be accessed via the following link: https://dhsprogram.com/data/available-datasets.cfm.

**Funding:** The author(s) received no specific funding for this work.

**Competing interests:** The authors have declared that no competing interests exist.

**Abbreviations:** ANC, Antenatal Care; CSA, Central Statistics Agenc; EA, Enumeration area; EDHS, Ethiopian Demographic and Health Survey; FNAC, Focused Antenatal Care; LBW, Low Birth Weight; WHO, World Health Organization.

wealthy households (IRR = 1.12, 95% CI: 1.051, 1.189). In the zero-inflated model, educational status, household wealth index, region, and religion show significant association with antenatal care service utilization uptake becomes zero.

## Conclusion

In rural Ethiopia, the rate of antenatal care service utilization has been lower than the respective current national statistics. A significant proportion of mothers who received antenatal care did not receive enough visits. Living in developing regions, following the Orthodox faith, being educated, and having a better home wealth position reduces the likelihood of skipping antenatal treatment.

## Introduction

The World Health Organization (WHO) envisions a world where every expecting woman and infant receives exceptional care during their pregnancy, delivery, and postpartum period [1]. The term "antenatal care" (ANC) refers to the treatment that qualified medical professionals give to teenage girls and expectant mothers in order to promote the healthiest outcomes for both the mother and the unborn child [2]. ANC lowers the morbidity and death rates for mothers and perinatals [3].

Maternal death rates are too high, according to the WHO. In 2020, pregnancy and the postpartum period claimed the lives of almost 287,000 women. While the majority of maternal fatalities in 2020 might have been avoided, over 95% of them happened in low- and middle-income nations. About 70% of maternal fatalities (202,000) occurred in Sub-Saharan Africa alone [4].

The 2016 Ethiopian Demographic and Health Survey found that Ethiopia had a maternal mortality ratio of 412 deaths for every 100,000 live births. Sixty-two percent of women obtained ANC from a qualified provider. In their most recent pregnancy, 32% of women had at least four ANC visits. 26% of deliveries were made in institutions. Low birth weight (LBW) was present in 13% of the babies [5].

Ethiopia is implementing the WHO's Focused Antenatal Care (FANC) strategy across all healthcare facilities nationwide. Pregnant women will receive ANC services four times during their pregnancy using the targeted FANC approach. Routine ANC appointments are scheduled for low-risk expectant mothers at eight to twelve weeks, twenty-four to twenty-six weeks, thirty-two weeks, and thirty-three to thirty-eight weeks of pregnancy. Women facing high-risk pregnancies and complications will require additional ANC visits [6–8]. Tetanus toxoid injections, iron and folate supplements, deworming drugs, blood tests for anemia and infection screening, urine testing, and a thorough physical examination are among the services provided [9].

Low prenatal care attendance has been linked to an increased risk of adverse pregnancy outcomes, according to several research [10]. The likelihood of stillbirth and premature delivery rose as the number of ANC visits decreased [11]. It was also discovered that women who had an ANC visit fewer than four times throughout their pregnancy and those who had no ANC follow-up had LBW babies [12].

Various factors related to the situation and sociodemographic attributes impact the number of ANC services received. Regarding sociodemographic characteristics, the frequency of prenatal care visits was found to be positively associated with advanced age, higher income, and

increased educational attainment. In contrast, factors such as parity, gestational age at delivery, the timing of ANC initiation, and health issues during pregnancy were all associated with a lower number of ANC visits. Factors such as how clients are received, service waiting times, and proximity to health facilities were associated with the utilization of ANC services. Additionally, the utilization of ANC services was connected to women's comprehension of ANC, living conditions, and access to electricity at home [13–21]. The number of ANC visits was influenced by factors such as the age of the woman, her financial situation, her educational level, the educational status of her spouse, her autonomy in making healthcare decisions, and her birth order [15].

The Ethiopian Demography and Health Survey (EDHS)-2016, which was examined using a logistic regression model, is the basis for several research conducted in Ethiopia. However, because the outcome variable is counted, information is lost as a result. Furthermore, there is a dearth of current national representative data. Because of this, we use recent national data in this study and take into consideration the methodological constraints of earlier research by using the proper technique of analysis for count data. The purpose of this study was to examine the frequency of ANC visits and the factors that influence them among rural Ethiopian women who had birth during the five years before to the survey.

Given that the majority of Ethiopia is rural, information on the prevalence of prenatal care usage and related characteristics would be crucial in forming policies and supporting decisions at different governmental levels.

## Methods and materials

### Study design, setting, and period

Ethiopia, which falls between 3˚ and 15˚ North latitude and 33˚ 48˚ East longitude, is in the northeastern region of Africa, where the study was carried out. Ethiopia is split administratively into eleven geographical areas. Every area is further divided into zones, which are further divided into woredas, towns, and kebeles within each towns. Additionally, Kebele is split into handy census enumeration areas (EAs). The majority of the health and demographic indicators were intended to be estimated for each of the 11 areas using the sample design.

The Federal Ministry of Health, the Ethiopian Public Health Institute, and the Central Statistical Agency worked together to collect the intermediate EDHS 2019 data set, which was used in this study. Data were obtained by contacting through personal accounts at https://www.dhsprogram.com, after providing justification for the request.

The sampling frame used for the 2019 EMDHS was a frame of all census EAs created for the 2019 Ethiopia Population and Housing Census (EPHC) and conducted by the Central Statistical Agency (CSA). Data collection took over 3 months, from March 21 to June 28, 2019.

### Source and study population

The source population consisted of all women in Ethiopian rural areas who are of reproductive age (15–49). The study population consisted of rural women selected from the designated enumeration areas who had given birth during the five years prior to each survey. Women without birth certificates or who reside in metropolitan areas were not included in the research.

### Sample size and sampling procedure

Two steps of stratification and selection went into selecting the intermediate EDHS 2019 sample. 21 sample strata were produced by first stratifying according to region and then further stratifying each area as urban and rural. In each sample stratum, 305 EAs (94 urban and 211

rural) were chosen using a probability proportionate to their size. Using a systematic random sample technique, homes were proportionately chosen from each EA for the second stage. For the purpose of this study's analysis, 2970 women were used.

To maintain the representativeness of the sample data, weighted values were calculated using children's records (KR) EDHS datasets. A total of 2,896.7 women were enrolled in the study (**Fig 1**).

### Study variable

The number of ANC visits during the last pregnancy is the study's outcome variable. Prenatal care was counted as a number between 0 and 20, however we recoded the data as 0 up to 7 visits and included the remaining visits as ≥8 visits. Women's age, religion, marital status at the time, educational attainment, household wealth index, geography, number of living children, and birth order are among the independent factors. The areas were divided into three categories: developed (Tigray, Amhara, Oromia, and South Nations Nationalities people), developing (Afar, Somali, Gambella, and Benishangul Gumz), and metropolitan (Harar and Dire Dawa) [22, 23].

### Method of data analysis

Many researchers have theoretical questions that involve count variables. A count variable is a variable that takes on discrete values (0, 1, 2…) reflecting the number of occurrences of an event in a fixed time. A count variable can only take on positive integer values or zero because an event cannot occur a negative number of times [24]. Counting data in many studies may have an excess of zeros. In the same process as other positive counts, zero counts may not occur.

### Poisson regression model

Count result data may be used to investigate the relationship between many factors using regression modeling approaches. However, because the result variable does not satisfy the transition from negative infinity to positive, conventional linear regression is typically not feasible for count data. Since count data on uncommon events or occurrences are nearly always skewed and non-normal, traditional methods that rely on normalcy are insufficient.

In contrast to traditional linear regression, the Poisson regression model describes the change in terms of log values, which have no bearing on the raw count values.

Most modern thinking in the topic starts with the Poisson regression model since the dependent variable (number of ANC visits) is a non-negative integer. The likelihood that a pregnant woman $i$ would see $yi$ antenatal care service visits during her nine months of pregnancy (where $yi$ is a non-negative integer) is determined by the following in a conventional Poisson regression model [25]:

$$(yi) = \frac{e^{(-\mu i)} \mu i^{(yi)}}{yi} \quad yi = 0, 1, 2, \ldots (\mu i > 0), \tag{1}$$

Where μi is the Poisson parameter for pregnant women i, which is equal to the expected number of antenatal care service visits in nine months, E(yi), and p(yi) is the probability that pregnant women entity i will have yi antenatal care service visits in the nine months of pregnancy period.

### Negative binomial regression model

A modification of the Poisson model that takes into account potential data over-dispersion is the negative binomial, or Poisson-gamma, model. The Poisson parameter in this model is

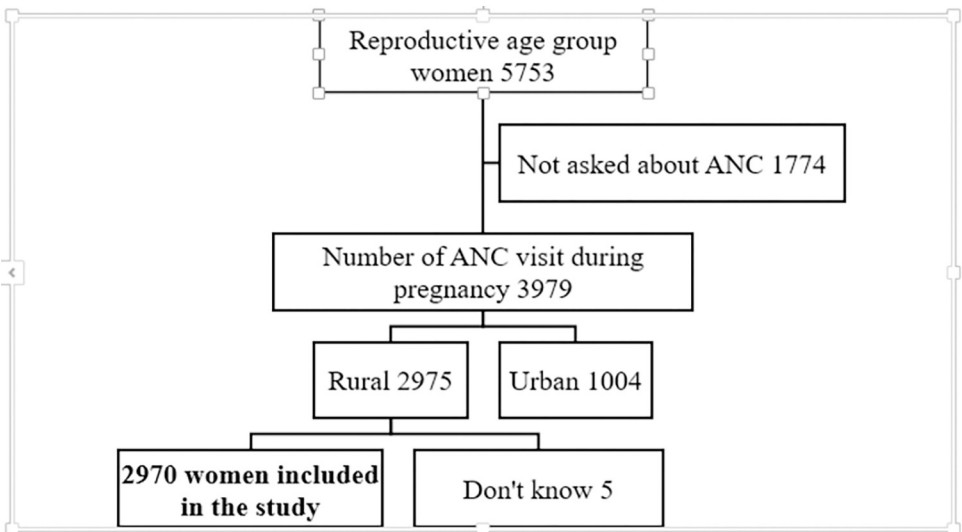

**Fig 1. Sampling and exclusion procedures to identify the final sample size in intermediate 2019 EDHS.**

assumed to have a gamma probability distribution. The Poisson parameter for each observation i is rewritten as $\mu i = \exp(\beta Xi + \varepsilon i)$, where exp ($\varepsilon i$) is a gamma-distributed error term with mean 1 and variance K. This yields the negative binomial model. This term's inclusion permits the variance to deviate from the mean in the following ways:

$$\text{Var}[y_i] = E[y_i][1 + K[y_i]] + E[y_i] + KE[y_i] \tag{2}$$

For the negative binomial distribution, the probability mass function is as follows:

$$p(yi) = \binom{yi + yi\, r - 1}{yi} p^r (1 - p)^{yi},\ r = 0, 1, 2, 3, \ldots \tag{3}$$

The likelihood of success in each trial is represented by the parameter p, which is computed as $p = r/\mu i + r$. The mean of the observations is denoted by $\mu i = \exp(y)$, and r represents the inverse of the dispersion parameter, or $r = 1/k$ [26].

The choice between these two models depends on the value of 'k', as the Poisson regression model limits the negative binomial regression model as 'k' approaches zero. The over-dispersion parameter is a common term used to describe parameter k. The negative binomial model may not be adaptable enough to handle situations where there are extra zeros, even though it can address an over-dispersion problem. In these situations, zero-inflated models be used to tackle the issue.

## Zero-inflated models

The negative binomial model is unable to sufficiently describe the over-dispersion that results from situations when the preponderance of zero counts is a significant contributing factor. In these situations, a zero-inflated Poisson or zero-inflated negative binomial model can be used to fit the data [27].

## Zero-inflated Poisson (ZIP) regression model

Based on the idea that a conventional count structure cannot handle the extra zero density, the ZIP regression model functions. A binary logit or probit model that takes into consideration

the possibility of an ANC visit entity being in zero or non-zero states determines a splitting regime that models a woman who did not visit for ANC vs a woman who visited for ANC throughout her pregnancy [24]. The result is always a zero count in one regime (R1), and a regular Poisson process for the counts in the other domain (R2).

Suppose that:

$p[y_i \varepsilon R_1] = \acute{\omega}_i$; $p[y_i \varepsilon R_2] = (1 - \acute{\omega}_i)$; i = 1, 2, . . ., n. Where $\acute{\omega}_i$ is inflation probability.

Thus, the occurrence of $Y_i$ follows the following distributions:

$$(Yi = yi) = \begin{cases} (1 - i)\, e^{-\mu i}\mu i^{yi}/yi! & \text{when yi} > 0 \\ i + (1 - i)\, e^{-\mu i} & \text{when yi} > 0 \end{cases} \quad \mu i > 0 \text{ and } 0 \leq i < 1. \tag{4}$$

The mean and variance of ZIP distribution are $\exp(y_i) = (1 - \acute{\omega}_i)\,\mu_i = \mu_i$ and $\text{var}(y_i) = \mu_i + (\acute{\omega}_i / 1 + \acute{\omega}_i)\,\mu_i^2 = (1 + \acute{\omega}_i)(\mu_i + \acute{\omega}_i\mu_i^2)$ indicating that the marginal distribution of $y_i$ exhibits over-dispersion of the data if ($\acute{\omega}_i > 0$). This reduces to the standard Poisson model when $\acute{\omega}_i = 0$.

## Zero-inflated negative binomial (ZINB) regression model

Two different mechanisms for generating data are assumed by the ZINB regression model. Similar to the ZIP distribution, the ZINB distribution is a mixed distribution with a probability of $\acute{\omega}i$ for excess zeros and a probability of $(1 - \acute{\omega}i)$ for the remaining counts, which follows a negative binomial distribution. Since the negative binomial distribution is a combination of Poisson distributions, over-dispersion may be approximated by allowing the Poisson, mean $\mu$, to be distributed as gamma. The ZINB distribution is given by:

$$P(Yi = yi) = \begin{cases} (1 - i)\Gamma(yi + \tau)/yi!\Gamma(\tau)(1 + \mu i/\tau) - \tau(1 + \mu i/\tau); & \text{when yi} = 1, 2, \dots \\ i + (1 - i)(1 + \mu i/\tau) - \tau\,; & \text{when yi} = 0 \end{cases} \tag{5}$$

The mean and variance of the ZINB distribution are $E(Y_i) = (1 - \acute{\omega}_i)\,\mu_i$ and $\text{Var}(Y_i) = (1 - \acute{\omega}_i)\mu_i\,(1 - \acute{\omega}_i\mu_i + \mu_i/\tau)$, respectively. It is to be noted that this distribution approaches the ZIP distribution and the negative binomial distribution as $\tau \to \infty$ and $\acute{\omega}_i \to 0$, respectively. If both $1/\tau$ and $\acute{\omega}_i \approx 0$ then the ZINB distribution reduces to the Poisson distribution [26, 28].

## Data processing and analysis

Data cleansing was done to make sure it was compatible with the EMDHS 2019 descriptive report. Re-coding, variable creation, labeling, and analysis were all done with Stata 17. Since the dependent variable, the ANC follow-up, is a non-negative integer, the Poisson regression model was employed as a starting point if the mean and variance were equal. The results showed that the mean was 2.59 and the variance was 4.13. The assumption is therefore disproved. This suggests that the data are overly dispersed. A Poisson regression extension that produces precise findings must be taken into consideration in order to handle over-dispersion and excess zeros in the data, often known as zero-inflated Poisson models [29].

$$P(Yi = yi) = \begin{cases} \pi + (1 - \pi i)e^{(-\mu i)\mu}, & \text{if } yi = 0 \\ (1 - \pi i)e^{(-y\mu i!i)}\mu yi, & \text{if } yi = 1, 2, 3.. \end{cases} \quad 0 \leq \pi i \leq 1.$$

Both the count portion and the zero-inflated section were analyzed. Lastly, 95% confidence intervals for incidence rate ratios were displayed. P values $< 0.05$ were used to define statistical significance. We check the models further below.

## Check the fitness of the model

Over-dispersion in the data may be tested in Poisson regression analysis using Pearson Chi-square goodness of fit statistics. There will be over-dispersion in the data set if the Pearson Chi-square statistic to degrees of freedom ratio is more than one [30]. By this approach, the Pearson Chi-square statistic is 4788.248 with matching degrees of freedom of 2951 using multivariable conventional Poisson regression analysis. The ratio of these statistical magnitudes is 4788.248/ 2951 = 1.62258, which is higher than one and indicates the presence of over-dispersion in the data. By adopting the following: Ho = model fits well and Ha = model doesn't fit well, P-value = 0.05, we can also verify the Pearson Chi-square goodness of fit statistics. In this situation, we reject the null hypothesis, suggesting that the model does not fit well.

A statistical test of the hypothesis: $H_0$: $\alpha = 0$ and $H_1$: $\alpha > 0$. If the P-value of LRT <0.05, then there is an over-dispersion and the Negative Binomial model is chosen. This is a commonly used method of determining whether the Negative Binomial regression model has over-dispersion. Where $\alpha$, also referred to as Stata's alpha, is an auxiliary parameter.

Following the multivariable negative binomial regression analysis, the hypothesis was rejected and the Stata alpha value was 0.3420119 with a p-value of 0.000, both of which are statistically significant. This suggested that the data set contained over-dispersion as well.

## Deviance test

Deviance is commonly used in statistical hypothesis testing and is a goodness-of-fit measure for a statistical model in statistics. It is a generalization of the idea of using the sum of squares of residuals in traditional least squares to scenarios where the model is fitted using maximum likelihood. For exponential dispersion models and extended linear models, it is essential. It is defined by -2*log-likelihood (-2LL). In terms of model comparison, the model with the lesser deviation is the superior model. By taking the differences in the deviation to obtain a χ2 value (i.e., χ2 = (-2LL1)-(-2LL2)), along with a corresponding p-value, you may ascertain if the difference in deviance is significant [31].

The (total) deviance for a model $M_0$ with estimates $\hat{\mu} = E[Y|\hat{\theta}o]$ based on a dataset y, may be constructed by its likelihood as,

$$D(y, \hat{\mu}) = 2(log[p(y|\hat{\theta}s)] - \log[p(y|\hat{\theta}o)])$$

In this case, $\hat{\theta}s$ stands for the saturated model's fitted parameters, and $\hat{\theta}o$ for the model $M_0$'s fitted parameter values. This formula is just the log-likelihood ratio of the entire model times two. When comparing two models, the deviation is employed [32].

To choose the analysis model, tests were run on Poisson regression, negative binomial regression, zero-inflated Poisson regression, and zero-inflated negative binomial regression models. We compare the models using the Deviance test (null and model), Log-likelihood (ll), the Akaike information criterion (AIC), and the Bayesian information criterion (BIC), and we show the results [33] (**Table 1**).

The provided analysis demonstrates thorough model evaluation using various metrics to assess fit, including Deviance, Log-likelihood, AIC, and BIC. The Deviance, AIC, and BIC values of the zero-inflated model version are marginally lower than those of the other models. The Zero-Inflated Poisson model shows the best fit among the models tested.

Using Zero-Inflated Poisson (ZIP) regression to analyze antenatal care utilization presents several challenges. The ZIP model assumes separate processes for zero counts and positive counts, which can impact accuracy if these assumptions do not align well with the data. The complexity of ZIP models can make interpretation difficult due to their dual-process structure,

**Table 1. A model comparison using deviance, log-likelihood, AIC, and BIC.**

| Model | Deviance(null) | ll (null) | Deviance(model) | ll (model) | AIC | BIC |
|---|---|---|---|---|---|---|
| poisson | 13,018.268 | -6509.134 | 12,156.64 | -6078.32 | 12194.64 | 12308.57 |
| nbreg | 12,225.552 | -6112. 776 | 11,806.24 | -5903.12 | 11846.24 | 11966.17 |
| **zip** | **10,570.786** | **-5285. 393** | **10,478.924** | **-5239.462** | **10554.92** | **10782.78** |
| zinb | 10,570.786 | -5285. 393 | 10,478.924 | -5239.462 | 10556.92 | 10790.78 |

and reliable results typically require a larger sample size, with smaller samples potentially affecting model stability. Additionally, selecting appropriate covariates and interactions is crucial to avoid misspecification.

## Ethics declarations

The study did not require ethics approval as it utilized secondary data that was publicly available. We accessed the data through the Demographic Health Survey web archive, where we registered and formally requested the datasets. Following this, permission was granted to view and download the data files. Throughout the research process, we adhered to all relevant guidelines and regulations to ensure compliance and integrity.

## Results

From weighted 2,896.7 pregnant women, 1085.55(37.47%) women use four or more antenatal care during their current pregnancy. About 848.38(29.29%) women do not attend antenatal care during pregnancy (**Table 2**).

## The magnitude of ANC utilization among pregnant women in rural Ethiopia, 2019

The highest percentage of ANC visits were observed from 25–29 age groups of women (29.65%), while the lowest visits were observed from above 44 age group women (2.25%).

The number of ANC visits increased significantly with educational attainment. Women with no education had the fewest visits, whereas women with greater education had the most. ANC visits were noted among a notably high number of Orthodox and Muslim religious adherents.

**Table 2. Number of women experiencing antenatal care in Ethiopia, 2019.**

| Number of ANC visit | Frequency | Percent |
|---|---|---|
| 0 | 848.38 | 29.29 |
| 1 | 110.38 | 3.81 |
| 2 | 248.05 | 8.56 |
| 3 | 604.31 | 20.86 |
| 4 | 645.61 | 22.29 |
| 5 | 249.8 | 8.62 |
| 6 | 127.6 | 4.40 |
| 7 | 25.52 | 0.88 |
| ≥8 | 37.02 | 1.28 |
| Mean | 2.586149 | |
| SD | 2.031962 | |

**Table 3. Number of antenatal care services utilization by socio-demographic characteristics of pregnant women in rural Ethiopia, 2019.**

| Variable | Weighted | | | Unweighted | | |
|---|---|---|---|---|---|---|
| | Number | Percent | Mean ± SD | Number | Percent | Mean ± SD |
| **AGE** | | | **3.5195±1.4182** | | | **3.4549±1.4259** |
| 15–19 | 174.898656 | 6.04 | | 203 | 6.84 | |
| 20–24 | 542.872892 | 18.74 | | 584 | 19.66 | |
| 25–29 | 858.849617 | 29.65 | | 887 | 29.87 | |
| 30–34 | 583.05244 | 20.13 | | 593 | 19.97 | |
| 35–39 | 461.846759 | 15.94 | | 431 | 14.51 | |
| 40–44 | 209.892436 | 7.25 | | 202 | 6.80 | |
| 45–49 | 65.2649793 | 2.25 | | 70 | 2.36 | |
| **MARITAL STATUS** | | | **1.9413±0.2351** | | | **1.9269±0.2603** |
| Not-married | 169.96328 | 5.87 | | 217 | 7.31 | |
| Married | 2,726.7145 | 94.13 | | 2753 | 92.69 | |
| **EDUCATION** | | | **0.4797±0.6225** | | | **0.4788±0.6419** |
| No education | 1,706.5577 | 58.91 | | 1789 | 60.24 | |
| Primary | 990.5694127 | 34.20 | | 940 | 31.65 | |
| Secondary & above | 199.550714 | 6.89 | | 241 | 8.11 | |
| **RELIGION** | | | **1.9437±0.8373** | | | **1.9471±0.7696** |
| Orthodox | 1,041.8351 | 35.97 | | 893 | 30.07 | |
| Muslim | 1,035.4217 | 35.75 | | 1410 | 47.47 | |
| Protestant | 760.049514 | 26.24 | | 598 | 20.13 | |
| Other | 59.3714649 | 2.05 | | 69 | 2.31 | |
| **REGION** | | | **2.0733±0.2815** | | | **2.2997±0.6372** |
| Metropolitans | 16.3263018 | 0.56 | | 291 | 9.8 | |
| Developed | 2,651.6356 | 91.54 | | 1498 | 50.44 | |
| Developing | 228.71592 | 7.90 | | 1181 | 39.76 | |
| **WEALTH INDEX** | | | **1.7068±0.8199** | | | **1.6172±0.8166** |
| Poor | 1,522.2859 | 52.55 | | 1776 | 59.80 | |
| Middle | 701.327108 | 24.21 | | 555 | 18.69 | |
| Rich | 673.064816 | 23.24 | | 639 | 21.52 | |
| **NO. OF CHILDREN** | | | **1.9424±0.8693** | | | **1.9502±0.8676** |
| 0 | 28.9077196 | 1.00 | | 40 | 1.35 | |
| 1–2 | 1,095.6276 | 37.82 | | 1075 | 36.20 | |
| 3–4 | 785.535901 | 27.12 | | 848 | 28.55 | |
| ≥5 | 986.606511 | 34.06 | | 1007 | 33.91 | |
| **BIRTH ORDER** | | | **2.4559±0.7902** | | | **2.4734±0.7835** |
| 1st | 544.707017 | 18.80 | | 541 | 18.22 | |
| 2nd | 486.666834 | 16.80 | | 482 | 16.23 | |
| 3rd & above | 1,865.3039 | 64.39 | | 1947 | 65.56 | |

Women who resided in developing regions and those who lived in the poorest areas had the lowest numbers of ANC visits, respectively. More than half of ANC visits are in the developed region part of Ethiopia.

When we compare with marital status number of ANC visits is higher in married women. The number of ANC visits are higher in women who are having their 3rd and above children (**Table 3**).

## Factors associated with the frequency of antenatal care

According to the Zero-Inflated Poisson model, the frequency of antenatal care utilization is significantly correlated with religious beliefs, household wealth index, and educational status.

In comparison to women without formal education, the frequency of prenatal visits was 1.1 (IRR = 1.1, 95% CI: 1.0426, 1.1754) and 1.2(IRR = 1.2, 95% CI: 1.0932, 1.3089) times higher among women enrolled in primary, and higher education, respectively.

The number of ANC visits was 13% (IRR = 0.87, 95% CI: 0.813, 0.9354) and 31% (IRR = 0.69, 95% CI: 0.5521, 0.8701) times more among Orthodox followers than Protestant and other religious followers, respectively.

Women in wealthy households were substantially more likely (IRR = 1.12, 95% CI: 1.0515, 1.1897) to receive prenatal care than women in impoverished households.

Educational attainment, family wealth index, geography, and religion have a strong correlation with the zero-inflated model's zero prenatal care service uptakes.

The number of antenatal care was 60% and 90% less likely to become zero among women who attend primary (AOR = 0.40, 95% CI: -1.1422, -0.6562) and secondary & higher (AOR = 0.10, 95% CI: -3.0741, -1.4181) education than women with no education.

As compared to Orthodox religious adherents, the percentage of prenatal care was 2.17, 2.41 and 3.23 more likely to become zero among Protestants (AOR = 2.17, 95% CI: 0.4668, 1.0848), Muslims (AOR = 2.41, 95% CI: 0.6026, 1.1618) and other religions (AOR = 3.23, 95% CI: 0.5125, 1.8343) respectively.

The likelihood that prenatal care for women residing in Harar and Dire Dawa would become zero was 1.7% higher than for women living in developing regions (AOR = 1.7, 95% CI: 0.1937, 0.8758).

The number of antenatal care was 46.16% and 66.76% less likely to become zero among women living in middle-class households (AOR = 0.53, 95% CI: -0.8963, -0.3411) and wealthy households (AOR = 0.33, 95% CI: -1.4168, 0.7859) (**Table 4**).

## Discussion

This study uncovers critical gaps in antenatal care utilization among rural Ethiopian women, showing that while ANC attendance has increased, many still receive insufficient care. Using recent national data and the Zero-Inflated Poisson model, which better handles excess zeros and high variability in ANC visits, the study provides a more accurate depiction of ANC patterns. This approach addresses methodological constraints of earlier research and highlights the urgent need for targeted public health interventions and policies to improve prenatal care services and address disparities.

In this study, 70.71% of women attended antenatal care at least once during their current pregnancy. The finding is consistent with studies in Afghanistan (69.3%) [34], Zambia (69%) [35], and Nepal (76.0%) [36]. The finding is higher than the study in the rural part of Ethiopia (2016 EDHS) (58.8%) [15], Benishangul Gumuz Region, Ethiopia (37.7%) [37], Nigeria (65.1%) [38], and Eastern Ethiopia (53.6%) [38]. The finding of the present study was lower than studies in Southern Ethiopia (76.2%) [39], Southwestern Ethiopia (91.9%) [40], Ghana (98.3%) [41], Pakistan (83.5%) [42], and Guinea (80.3%) [19]. Compared with a study done using 2016 Ethiopian DHS data, the number of at least one ANC visit in rural parts of Ethiopia has increased by 11.91%. This suggests that the country (Ethiopia) is working to increase ANC coverage, focusing on particular areas.

Only 37.47% of the women in this study receive the required amount of prenatal care as per the recommendation of WHO. The results of this study are less than the results of Pakistan (57.3%) [42], Nigeria (56.2%) [38], Rwanda (54%) [43], India (51.7%) [44], and sub-Saharan

**Table 4. Factors associated with ANC service utilization among pregnant women in Ethiopia, 2019.**

| Variables | IRR | [95% CI] | | Inflated part AOR | [95% CI] | |
|---|---|---|---|---|---|---|
| | | Lower bound | Upper bound | | Upper bound | Lower bound |
| **AGE** | | | | | | |
| **15–19** | 1 | | | | | |
| **20–24** | 1.063927 | .9471685 | 1.195079 | 0.993911 | -.469237 | .4570222 |
| **25–29** | 1.070986 | .9468209 | 1.211434 | 0.66123 | -.9126337 | .0853255 |
| **30–34** | 1.134167 | .9905992 | 1.298543 | 0.66499 | -.9473397 | .1313713 |
| **35–39** | 1.121547 | .9686153 | 1.298624 | 0.73984 | -.8667741 | .2641394 |
| **40–44** | 1.156315 | .9437282 | 1.485622 | 0.91621 | -.7050841 | .5300574 |
| **45–49** | 1.184071 | .9437282 | 1.485622 | 1.27104 | -.5001866 | .9798563 |
| **EDUCATION** | | | | | | |
| **No education** | 1 | | | | | |
| **Primary** | 1.106979 * | 1.042564 | 1.175374 | 0.40688** | -1.142228 | -1.142228 |
| **Secondary & above** | 1.196179 * | 1.093182 | 1.308882 | 0.10581** | -3.074109 | -3.074109 |
| **MARITAL STATUS** | | | | | | |
| **Not married** | 1 | | | | | |
| **Married** | .9980148 | .9058107 | 1.099604 | 1.402197 | -.3554476 | .4230555 |
| **RELIGION** | | | | | | |
| **Orthodox** | 1 | | | | | |
| **Protestant** | .8720591* | .8130246 | .9353802 | 2.172254** | .4667755 | 1.084756 |
| **Muslim** | .9889777 | .9257279 | 1.056549 | 2.41621** | .6025562 | 1.161846 |
| **Other** | .6930903* | .5520769 | .8701218 | 3.23299** | .5124985 | 1.834314 |
| **WEALTH INDEX** | | | | | | |
| **Poor** | 1 | | | | | |
| **Middle** | 1.065167 | .9981475 | 1.136687 | 0.538462** | -.8963542 | -.3411387 |
| **Rich** | 1.118434* | 1.051456 | 1.189679 | 0.332423** | -1.416825 | -.785869 |
| **REGION** | | | | | | |
| **Metropolitan** | 1 | | | | | |
| **Developed** | 1.068603 | .9711252 | 1.175864 | 1.187854 | -.2024938 | .5467934 |
| **Developing** | .9900711 | .9022144 | 1.086483 | 1.707038** | .1936818 | .8758375 |
| **NO. OF CHILDREN** | | | | | | |
| **0** | 1 | | | | | |
| **1–2** | 1.195756 | .9226929 | 1.549629 | .68267 | -1.228773 | .4652851 |
| **3–4** | 1.163943 | .866604 | 1.563302 | .076376 | -1.238818 | .7244 |
| **>5** | 1.150852 | .8545697 | 1.549858 | .767604 | -1.252697 | .7237365 |
| **BIRTH ORDER** | | | | | | |
| **1st** | 1 | | | | | |
| **2nd** | .9526271 | .875972 | 1.03599 | 1.04653 | -.3364824 | .427452 |
| **3rd & above** | .9851273 | .8465988 | 1.146323 | 1.177587 | -.4417389 | .768674 |

*Has significant association in IRR and ** on AOR

Africa (58.53%) [21]. The results are greater than those obtained in Eastern Ethiopia (15.3%) [45], Bangladesh (32%) [46], the rural areas of Ethiopia (2016 EDHS) (27.3%) [15], and Zambia (29%) [35]. The study population coverage and study environment (being a rural area, there may be limited infrastructure, health facility coverage, and awareness regarding ANC follow-up) could be the cause of the discrepancy. Additionally, it demonstrated over the 10.17% increment in the 2016 statistics. This increment may be due to initiatives to raise public

awareness, health promotion campaigns, and the growth of medical facilities and providers in rural areas.

When women's educational status rises, so does the frequency of prenatal care. Increasing educational status increases the likelihood of using prenatal care services. The finding was supported by previous research in Western Ethiopia [37], Ethiopia [20], rural Ethiopia (2016 EDHS) [15], Kenya [47], Guinea [19, 48], Afghanistan [34], Angola [18], East African countries [49], India [50], Ghana [51], Nepal [52], Nigeria [53], and Sub-Saharan Africa countries [54]. Educated women should be empowered to get services [55, 56], and education allows women to decide on their health-seeking behavior [57, 58]. Moreover, educated women know of the danger signs of pregnancy [59]. Educated women may be aware of the benefits of prenatal care and the various services for both the fetus and the woman [17, 60]. This suggests that the women's level of education raises their knowledge of the importance of getting regular prenatal care appointments to get health services.

Prenatal care usage varied according to religion. Orthodox Christians use prenatal care at a higher rate than Protestants or other religious followers. Research indicates that there are differences in religion when it comes to the use and frequency of prenatal treatment [13, 61, 62]. Further qualitative research might be needed to have more information on the effect of religion on the utilization of ANC.

As the household wealth index grows, the frequency of antenatal care visits increases. The outcome was in line with earlier research conducted in Bangladesh [46, 63], Nigeria [38], India [44], Ghana [41], Guinea [19], Angola [18], Nepal [64], Western Ethiopia [37], and Sub-Saharan Africa [54]. It could be because, in comparison to impoverished women, wealthy women typically have greater educational levels [65, 66], access to mass media [54, 67], and able to pay for frequent ANC visits [68].

Since the output of the study included count data, the model took into account various levels of analysis, and the results were more representative than other studies. Despite this advantage, the results could be subject to recall bias because some factors were overlooked. After all, the data set was intermediate and the data were taken from an event's past. Paternal variables, prior exposure variables, and quality-related data were excluded from the data set due to their nature.

The study reveals a need for targeted interventions to enhance prenatal care in rural areas. While attendance has risen, many women still lack adequate care. Policymakers should implement more effective public health campaigns, expand medical access, and address disparities in education, wealth, and religion to ensure comprehensive care for all women.

## Conclusion

The level of antenatal care visits among women who gave birth within the last five years before the survey was low in rural Ethiopia. The ANC service utilization rate in rural Ethiopia was lower than the national figures. The majority of the mothers who attend ANC did not receive the adequate number of visits recommended by the World Health Organization. Furthermore, maternal education, religion of mothers, and wealth index were major predictors of ANC service utilization. Being educated, having improved household wealth status, living in developed regions, and following the Orthodox religion decrease the probability of not attending antenatal care.

In this study, through simulation experiments, it was found that the zero-inflated model is better fitted to the data than the Poisson, negative binomial, and zero-inflated negative binomial regression models. Deviance, AIC, and BIC tests show that the Zero-inflated model was better fitted to the data which is characterized by excess zeros and high variability in the non-zero outcome.

## Acknowledgments

The authors would like to acknowledge the Ethiopian Central Statistical Agency for providing an authorization letter to access EDHS-2019 datasets to conduct this study.

## Author Contributions

**Conceptualization:** Bisrat Misganaw Geremew, Daniel Gashaneh Belay, Habtu Kifle Negash.

**Data curation:** Yitbarek Fantahun Mariye, Habtu Kifle Negash.

**Formal analysis:** Bisrat Misganaw Geremew, Habtu Kifle Negash.

**Methodology:** Bisrat Misganaw Geremew, Daniel Gashaneh Belay, Hiwot Tezera Endale, Habtu Kifle Negash.

**Software:** Bisrat Misganaw Geremew, Habtu Kifle Negash.

**Visualization:** Habtu Kifle Negash.

**Writing – original draft:** Bisrat Misganaw Geremew, Habtu Kifle Negash.

**Writing – review & editing:** Bisrat Misganaw Geremew, Yitbarek Fantahun Mariye, Hiwot Tezera Endale, Fana Kinfe Gebreegziabher, Habtu Kifle Negash.

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
