## [Decision Letter · Decision Letter 0]

16 Jul 2024

PONE-D-24-06799Number of antenatal care utilization and associated factors among pregnant women in rural Ethiopia: Zero-inflated Poisson regression of 2019 intermediate Ethiopian Demography Health SurveyPLOS ONE

Dear Dr. Negash,

Thank you for submitting your manuscript to PLOS ONE. After careful consideration, we feel that it has merit but does not fully meet PLOS ONE’s publication criteria as it currently stands. Therefore, we invite you to submit a revised version of the manuscript that addresses the points raised during the review process.

**ACADEMIC EDITOR: **The discussion part requires restructuring and modification. In the current version it is narrow and does not properly discuss the study results. Suggested outline for the diascussion part:Discussion

1.1 Rationale of the study (why it was done)

1.1.1 Main findings of the study

1.1.2 What makes your study unique

1.1.3 What it adds to what we already know

1.2 Subject of the discussion

Comparison of your results with neighboring countries, with countries of the same

development levels (income), with developed high-income countries). Agreement and

disagreement with the studies compared

1.3 Sum up of the study, study strengths and limitations

1.5 Clinical implication

We look forward to receiving your revised manuscript.

Kind regards,

Gulzhanat Aimagambetova

Academic Editor

PLOS ONE

Journal Requirements:

2. Thank you for stating the following in your Competing Interests section: "No computing of interest"

5. Please ensure that you refer to Figure 1 in your text as, if accepted, production will need this reference to link the reader to the figure.

6. We note you have included a table to which you do not refer in the text of your manuscript. Please ensure that you refer to Table 2 in your text; if accepted, production will need this reference to link the reader to the Table.

Reviewers' comments:

Reviewer's Responses to Questions

**Comments to the Author**

1. Is the manuscript technically sound, and do the data support the conclusions?

Reviewer #1: Yes

Reviewer #2: Yes

Reviewer #3: Yes

Reviewer #4: Yes

2. Has the statistical analysis been performed appropriately and rigorously? 

Reviewer #1: Yes

Reviewer #2: I Don't Know

Reviewer #3: Yes

Reviewer #4: I Don't Know

3. Have the authors made all data underlying the findings in their manuscript fully available?

Reviewer #1: Yes

Reviewer #2: Yes

Reviewer #3: Yes

Reviewer #4: Yes

4. Is the manuscript presented in an intelligible fashion and written in standard English?

Reviewer #1: Yes

Reviewer #2: Yes

Reviewer #3: Yes

Reviewer #4: Yes

5. Review Comments to the Author

Reviewer #1: Dear author

Its a very well written and structured article

All the sections including abstract, methodology, results and discussion are very well described.

The results are displayed in detail and elaborate.

Reviewer #2: � Comments to Authors:

Abstract:

• please correct the number stated in line 42, written in bold in the following sentence (The highest percentage (29.65%) of antenatal care visits was observed from 25 to 2age.)

Title:

• The title of the study is clear, inclusive and precise to the study’s objectives and aims. The authors stated the study design distinctly.

Introduction:

• The introduction is well structured, clear, and straightforward. The authors had effectively introduced the concept of antenatal care in alignment with the WHO FANC strategy, and highlighted the importance of regular ANC in promoting the health of mothers and their newborns (live birth weight in specific).

• The rationale and aim of the study are well stated.

• The relevant statics sourced from the WHO and the 2016 Ethiopian Demographic and Health Survey by the authors have established a robust citation that contributes significant value.

Methods:

• The authors conducted a database study.

• The authors effectively described the geographical location and administrative divisions of Ethiopia, which provides more comprehension for the non-Ethiopian audience.

• The section provides a detailed overview of the study design, setting, data collection methods, and sample size determination. In addition, they included essential information, needed for further understanding of the analytical framework. However, mentioning some potential challenges in choosing method of analysis would be a qualitative addition.

• Concerning the data analysis, the authors have successfully mentioned the source of data and the justification for accessing it, which adds transparency to the research process.

Results:

• The authors conducted a thorough analysis of the results, supported by sufficient amount of data justifying the conclusion. Nevertheless, the analysis assumed a linear relationship between variables, overlooking potential confounding factors that could impact the accuracy of correlations. Therefore, further research may be warranted to validate these hypotheses.

• The authors sub-categorized the study variables, which provided a more inclusive perspective.

Discussion:

• In the discussion section, the authors have effectively compared their findings with those of similar studies and offered reasonable explanations for both the similarities and differences.

• Furthermore, the authors did highlight the study's limitations in a transparent discussion of these limitations.

• Further clinical application might be mentioned for policy makers, together with future direction and recommendation.

Conclusion:

• Correctly answered the research question.

• The authors were able to mention the most important research results in a concise and clear manner.

Reviewer #3: The study is well formated and quite informative on the level of health care through out the country, good analysis of the data of ANC depending on different variables including geography,religion,education and socioeconomy,make the study is more reflecting the culture in concern.

Reviewer #4: table on competing interest typographical error written as "no computing of interest"

ethics statement should be more robust. the NHS data that was used should be well spelt out

the result in the abstract and that in the method result and findings are different. likewise, the number of women studied were different. 2970 and 2896.7 were quoted. since no of pregnant women are whole numbers, the value of 2896.7 is not appropriate.

were the women interviewed to get the number of ANC visits they had or all data were taken from the NHS data. this is important as line 371-372 stated the factor of recall bias. if the data has been recorded, there should not be recall bias.

6. PLOS authors have the option to publish the peer review history of their article (what does this mean?). If published, this will include your full peer review and any attached files.

Reviewer #1: **Yes: **Saida Abrar

Reviewer #2: **Yes: **Bayan Al Omari

Reviewer #3: **Yes: **Mohsen M A Abdelhafez

Reviewer #4: No

---

## [Author Response · Author response to Decision Letter 0]

28 Aug 2024

First of all, I would like to acknowledge your collective concern, time, devotion, patience, and valuable comments. Your insights have greatly contributed to improving my understanding of the area.

Reviewer comment #1: 

It’s a very well written and structured article. All the sections including abstract, methodology, results and discussion are very well described. The results are displayed in detail and elaborate.

Author response: Thank you for your kind words and detailed feedback. I am pleased to hear that you found the article well-written and structured, with clear descriptions and detailed results. Your comments are greatly appreciated.

Reviewer comment #2: 

1. Please correct the number stated in line 42, written in bold in the following sentence (The highest percentage (29.65%) of antenatal care visits was observed from 25 to 2age.)

Author response: Thank you for pointing out the discrepancy. I have reviewed the result section and corrected the percentage. The revised statement now accurately reflects that the highest percentage (29.65%) of antenatal care visits was observed among women aged 25-29 years.

2. The section provides a detailed overview of the study design, setting, data collection methods, and ample size determination. In addition, they included essential information, needed for further understanding of the analytical framework. However, mentioning some potential challenges in choosing method of analysis would be a qualitative addition.

Author response: Thank you for your feedback. I have added a discussion of potential challenges associated with the chosen method of analysis to provide a more comprehensive overview. This addition addresses possible issues related to the analytical approach and enhances the understanding of its application in the study.

3. The authors conducted a thorough analysis of the results, supported by sufficient amount of data justifying the conclusion. Nevertheless, the analysis assumed a linear relationship between variables, overlooking potential confounding factors that could impact the accuracy of correlations. Therefore, further research may be warranted to validate these hypotheses.

Author response: Thank you for your insightful comment. You are correct that the analysis assumed linear relationships between variables, and potential confounding factors may impact the accuracy of the correlations. The 2019 Ethiopian Mini DHS data used in our study did not include some variables that could influence the outcome, which may have limited our analysis. We acknowledge this as a limitation of using the Mini DHS data. Further research incorporating a broader set of variables and exploring potential confounders would be valuable to validate and refine these hypotheses.

4. Further clinical application might be mentioned for policy makers, together with future direction and recommendation.

Author response: Thank you for your suggestion. I have thoroughly revised the manuscript to include further clinical applications for policymakers. Additionally, I have added on the last part of my discussion on future directions and recommendations to provide more detailed guidance for subsequent research and practice.

Reviewer comment #3: 

The study is well formatted and quite informative on the level of health care throughout the country, good analysis of the data of ANC depending on different variables including geography, religion, education and socio-economy, make the study is more reflecting the culture in concern.

Author response: Thank you for your thoughtful comments and positive feedback. I appreciate your recognition. Your insights are valuable and encouraging.

Reviewer comment #4: 

1. Table on competing interest typographical error written as "no computing of interest"

Author response: First of all thank you for your time and comments. The “Competing interests” is written as “The authors declare that they have no competing interests.” 

2. Ethics statement should be more robust. the NHS data that was used should be well spelt out

Author response: Thank you for your feedback. I have revised the ethics statement to provide a more comprehensive account. The study utilized secondary data from the National Health Survey (NHS), which is publicly available and does not require individual ethics approval for use. We accessed the data through the DHS (Demographic and Health Surveys) web archive by registering and obtaining permission to download the datasets. All relevant ethical guidelines and regulations for secondary data use were strictly followed to ensure the integrity of the research.

3. The result in the abstract and that in the method result and findings are different. Likewise, the number of women studied were different 2970 and 2896.7 were quoted. Since no of pregnant women are whole numbers, the value of 2896.7 is not appropriate.

Author response: Thank you for pointing out the discrepancies. The number of women included in the study, as shown in Figure 1, is indeed 2,970. The figure 2,896.7 refers to the weighted number of women, obtained using the DHS (Demographic Health Surveys) data. To ensure representativeness, we applied a weighting factor using the Stata command generate wt=v005/1000000. The resulting weighted value, 2,896.7, represents the statistical adjustment rather than an actual fraction of individuals. We have clarified this distinction by separately presenting the "weighted" and "unweighted" figures in the results section to avoid confusion.

4. Were the women interviewed to get the number of ANC visits they had or all data were taken from the NHS data. This is important as line 371-372 stated the factor of recall bias. If the data has been recorded, there should not be recall bias.

Author response: Thank you for raising this important point. The data for the study was sourced entirely from the DHS data, and the number of antenatal care (ANC) visits was not collected through direct interviews. According to the "Guide to DHS Statistics, DHS-7 (version 2)," the data includes:

Numerators:

The number of women who received antenatal care for their last birth, categorized by the number of visits (variable m14).

The number of women who received antenatal care for their last birth, categorized by the number of months pregnant at the time of the first visit (variable m13).

Denominator: The number of women who had a birth in the last five years (variable midx = 1).

In the analysis, we focused on women whose last birth occurred within the past five years (midx = 1), which helps in ensuring the data's relevance. Despite this, recall bias can still occur, particularly for women who may find it challenging to remember the exact number of ANC visits or the timing of their first visit, especially if their last pregnancy was several years ago. This potential recall bias was noted in the discussion to account for variations in memory accuracy over time.

---

## [Editor Report · Decision Letter 1]

17 Sep 2024

Number of antenatal care utilization and associated factors among pregnant women in rural Ethiopia: Zero-inflated Poisson regression of 2019 intermediate Ethiopian Demography Health Survey

PONE-D-24-06799R1

Dear Dr. Habtu Kifle Negash,

We’re pleased to inform you that your manuscript has been judged scientifically suitable for publication and will be formally accepted for publication once it meets all outstanding technical requirements.

Kind regards,

Gulzhanat Aimagambetova

Academic Editor

PLOS ONE
---

## [Editor Report · Acceptance letter]

20 Sep 2024

PONE-D-24-06799R1 

PLOS ONE

Dear Dr. Negash, 

I'm pleased to inform you that your manuscript has been deemed suitable for publication in PLOS ONE. Congratulations! Your manuscript is now being handed over to our production team.

Kind regards, 

on behalf of

Dr. Gulzhanat Aimagambetova 

Academic Editor

PLOS ONE